# The Usability of IT Systems in Document Management, Using the Example of the *ADPIECare Dorothea Documentation and Nurse Support System*

**DOI:** 10.3390/ijerph19148805

**Published:** 2022-07-20

**Authors:** Dorota Kilańska, Agnieszka Ogonowska, Barbara Librowska, Maja Kusiak, Michał Marczak, Remigiusz Kozlowski

**Affiliations:** 1Department of Coordinated Care, Medical University of Lodz, Kościuszki Street 4, 90-131 Lodz, Poland; barbara.librowska@umed.lodz.pl (B.L.); panimaja@yahoo.com (M.K.); 2Department of Management and Logistics in Healthcare, Medical University of Lodz, Lindleya Street 6, 90-131 Lodz, Poland; agnieszka.ogonowska@umed.lodz.pl (A.O.); michal.marczak@umed.lodz.pl (M.M.); 3Center of Security Technologies in Logistics, Faculty of Management, University of Lodz, Matejki Street 22/26, 90-237 Lodz, Poland; remigiusz.kozlowski@wz.uni.lodz.pl

**Keywords:** IT, ICNP^TM^, nursing documentation, care plans, usability

## Abstract

Background: In 2016, an IT system was developed at MUL for the documentation of nursing practice. Preparing nursing students for the implementation of eHealth solutions under simulated conditions is crucially important for achieving the digital competencies necessary for health care systems in the future. Scientific evidence demonstrates that the use of an IT system in clinical practice shortens the time required for the preparation of documentation, increases the safety of clinical decisions and provides data for analysis and for the creation of predictive models for the purposes of HB HTA. Methods: The system was created through the cooperation of an interprofessional team at the Medical University of Łódź. The ADPIECare system was implemented in 2016 at three universities in Poland, and in 2017 a study of its usability was conducted using a questionnaire made available by *Healthcare Information and Management Systems Society*, “Defining and Testing EMR Usability MASTER V2 Final” on 78 nurses—students of MA in Nursing at Medical University of Łódź. Findings: Over 50% of the surveyed nurses indicated the usability of the system for the “effectiveness of documentation” variable. The same group of respondents had a positive attitude towards patient care planning with the use of the assessed system. In the opinions of the examined parties, positive opinions predominated, such as, e.g., “the system is intuitive”, “the system facilitates work”, “all patient assessments are in one place”, and “the time needed for data entry would be shortened”.

## 1. Introduction

Electronic IT systems should collect all appropriate normalised data generated by health care employees, such as nurses and doctors, in order to improve the quality and effectiveness of services provided to the patients [1]. Unfortunately, electronic medical records (EMR) may currently not be used to study the independent contribution of nursing to health care outcomes [2], thus the use of nursing terminology in this project, which will enable not only care planning, but also obtaining medical history and describing the health care status of the patient [3]. This enables the unification of care and its standardisation [4,5]. Providing appropriate care requires the measurement of outcomes of the nursing process [6]. Moreover, the types of outcomes that the nurses are required to measure and manage are inextricably linked with the definition of nursing itself. [7,8]. An increasing number of scientific contributions indicate that the digitisation of nursing care provides a measurable value for the patient and for the system [9,10]. Studies also indicate the importance of digitisation to improve the processes of care [9,11]. Electronic medical records could also enhance the quality of health care [12]. Research indicates that advantages related to the use of systems far outweigh the inconveniences [13] and improve clinical practice [14]. Analyses and studies also demonstrate that the use of new technologies results in a significant reduction in errors [15,16]. What is important is the enhancement of the decision-making process in clinical practice [11,12,17]. Research indicates a significant role of new technologies in improving the effectiveness and efficacy of healthcare [12,15,18]. Data gathering is an issue which should be addressed in nursing. It was proven that the quality of predictive models improved when nursing data (in addition to medical data) were taken into account [19,20]. An important expectation for the digitisation process, apart from improving quality and the communication between doctors and nurses, is to reduce costs of the health care system [21]. The knowledge of how to accurately document a patient’s status may literally mean life or death.

Incorrectly maintained documentation may expose a medical facility to legal claims and to uncontrolled abuse. One of the most famous cases in the history of medicine which resulted in regulating the number of working hours of resident physicians is a case study concerning the lack of access to clinical documentation. Experiences indicate that the implementation of medical information systems frequently suffers from many difficulties [22,23]. The strategic objectives to be achieved by 2030 in the “Policy paper” in Poland include, among others, goal 3. The implementation of instruments improving the quality of provided health services and the efficiency of the health care system, which assumes, among other things, the improvement of systems for collecting and managing medical information, in conjunction with implementing e-health projects [24]. This is repeated in the “Healthy future” policy, whereas the “Strategic frameworks of the development of the health care system for the years 2021–2027, with a perspective up to 2030” indicates that one of the policies is the development of digital services in the health care system; Goal 1.2 (Quality) Improvement of safety and clinical efficacy of health services; development of payer mechanisms to pay for quality [25]. Quality has its dimension in patient safety. A measurement of safety in the form of sentinel events used in the United Kingdom shows 120 reports relating to e-health services. As many as 33% are events that concern the usability of the system. The usability of a system is a feature that depends primarily on the user interface and the clarity of documentation; operability refers to the free use of all functions of the application by the user, as opposed to the remaining attributes, where simple operation is required (the ISO 9126 standard) [26]. In accordance with the next standard ISO 9241-11 usability depends on effectiveness, efficiency and satisfaction [27]. These features depend on the user’s environment and on user’s emotions when achieving the established goals: effectiveness is the precision and completeness which users achieve specified goals with; efficiency is the relationship between precision and performance of the task and the resources incurred to achieve this goal; satisfaction is the positive emotion of the users which results from accepting the operation of the system. Usability results from the user’s feelings and thoughts after interacting with an IT system. The emotional state of a person has a key impact on the results of their work and on their willingness to use the system again. Usability refers not only to websites, software or electronic services, but to all electronic devices and interactive systems with which we come into contact. Additionally, it should be emphasised that usability, which describes the ease of use of a system, does not overlap with the term functionality responsible for its capabilities and the amount of available options [28]. Jakob Nielsen defined *usability* as a set of five elements: *learnability, efficiency, memorability, errors,* and *satisfaction*. In order to assess the usability of the system, at least five users are needed. As claimed by Nielsen, this serves the development of the system, instead of spending resources on unnecessary tests which would result in the same effect [29].

The Figure 1 presents the method of achieving the application’s usability. It is based on the ISO 9241-11 standard and it applies to interactive cycles of product improvement.

In the column on the left side, a user was presented, as was his/her task, the tool he/she was using and his/her environment. On the opposite side, in the upper right corner the goals of the user are provided, which he/she implements through his/her actions with satisfaction and in an effective and efficient manner (the area of usability). The entire process can also fail. The user will not achieve the intended goal, which will result in dissatisfaction and feelings of frustration [27,30].

System usability is of key importance in the context of nursing work. Medical care in a hospital is performed mainly by nurses and includes processes intended to improve a patient’s health status, including, in particular, before and after a medical intervention. Nurses visit patients 157 times during a twelve-hour shift [31], and are responsible for the daily monitoring and managing of health care provided to patients [32]. A nursing role includes immediate detection and intervention when the patient’s clinical condition changes [33,34]. Nurses form a supervision system used for the early detection of patient complications and problems and have the best possibility of initiating actions which minimise negative patient outcomes [35]. Studies confirm the significant engagement of nurses as advocates for high quality, patient-focused care and cost-effective health care [36]. In the source literature, the time spent by nurses on performing activities related to intermediate care (documenting, administering) ranges from 22% [37] to 43.2% [38]. The digitisation of processes performed by nurses reduces workload and enables increasing the time for providing patients with direct care [16,39,40]. This is demonstrated by studies which show that electronic records shorten the documentation time by more than 50%; the share of nursing time dedicated to documentation amounted to 15.8%, 10.6% on paper and 5.2% on computer [41]. The time saved by digitisation is a measurable benefit for the patient and for the system [9,42]. It should be noted that until 2030, Europe may have a shortage of over 4 million of medical personnel [43]. All over Europe, the discrepancy between the demand for health care and the availability of personnel and other resources is growing, just as the awareness that digital transformation is of key importance to fill this gap. There are no studies in Poland showing the importance of IT systems for nursing care; however, based on the above studies, we can expect that IT systems that receive a positive user assessment will be more willingly used for documentation. Thus, it may mean that Polish nurses, just as in other countries, will have more time for direct patient care. Nurses, who devote a significant part of their time to administrative work, need more time for care which provides patient value. In 2012, studies were conducted in which it was examined how new nurses assess their knowledge and skills in clinical conditions, compared to the perception of the same skills by nurse managers. They have demonstrated gaps in 13 out of 28 areas of knowledge and skills deemed to be key for the effectiveness of electronic healthcare records (EHR). Almost 90% of new/beginning nurses and 75% of nurse managers participated in EHR training at their work sites, but only 20% of new/beginning nurses and only 7% of nurse managers stated that EHR was part of their training programme in nursing school. Over 60% of nurse managers agreed that starting nurses need more than 2 months in order to be competent with EHR use [44]. Providing students with an EMR training programme under simulated conditions, where they can train various scenarios and become skilled in a safe, supervised environment, is key for the digitisation process. This process is understood as increasing safe decision making based on actual, unequivocal and undeniable data. Most medical errors do not occur as a result of the incompetence or recklessness of nurses and other medical personnel. They occur because of defective systems and fragmented processes. The main culprit is defective documentation [45]. Appropriate and accurate documentation is necessary in order to avoid various types of errors in making clinical decisions by a doctor or a nurse, and to help to avoid the death of patients and the medical facility’s liability. In this area, EMR comes to aid. The effective training of nursing students in EMR may play a significant role [46].

### Characteristics of the ADPIECare Dorothea IT System

This application reflects all the process elements using the *ICNP^TM^* reference terminology. The first stage is the assessment of patient knowledge (Assessing Knowledge Of Disease—10030639), then the Diagnosis And Outcome (10016446), Care Planning (10035915) Implementing (10009840) and afterwards Evaluating (10007066), at the last stage showing the result in the form of a final diagnosis. In 2001, work commenced on the implementation of terminology recommended by the international nursing community and the UN, World Health Organisation—*International Classification for Nursing Practice—ICNP*. The terminology contains terms necessary to describe a nurse’s work, which are combined into blocks in accordance with the ISO 18104:2003 standard [47]. The United Nations recommends the use of *ICNP* terminology to gather statistics in nursing [48]. *ICNP* reflects the boundaries of nursing practice, and thus overlaps with other health terminologies such as *SNOMED CT*, simultaneously exceeding their scope. It is more comprehensive and more detailed than the Classification for Clinical Practice and has higher international use since it is based on the *OWL* ontology. Thus, when planning IT solutions, the aforementioned recommendations should be used. In 2011, based on this terminology, a recommendation by the Nursing eHealth Council of The Center Healthcare Information Systems Ministry of Health and *ICN* Accredited Centre of *ICNP* MUL was created, establishing the scope of data for nursing documentation and the use of structured nursing practice terminology. In 2015, the Minister for Health accepted the Recommendation of 11 September 2013 [49]. As a result, works commenced on the transposition of the Recommendation to technical language of *HL 7 CDA* (*HL 7* (Health Level Seven) *CDA* (Clinical Documentation). Health Level Seven or *HL7* refers to a set of international standards for the transmission of administrative and clinical data between applications used by various service providers)—an interoperability technical standard. This standard enables the exchange of data between units that provide care. This document was published in Polish National Implementation (PIK) and became a benchmark for the creation of IT systems for nursing documentation [48,50].

The undertaken implementation actions resulted in the **“***ADPIECare Dorothea documentation and nurse support system”,* which the first Polish system enabling the documentation of nursing work and teaching with the use of the international ICNP terminology.

The acronym ADPIECare reflects the care process shown in Figure 2—**A**ssessment/**D**iagnosis/**P**lanning/**I**nterventions/**E**valuation **Care**. The related work commenced on 7 March 2016 and ended on 8 August 2016. 

The ADPIECare system is an expert system which supports the work of an operator by suggesting diagnoses and activities. It does not make any decisions for the operator.

The functionality of the nursing works documentation system is a user panel which contains a special education module.

In the education module, the system presents the patient documentation (maintained by students). This allows the student’s tutor to track the assessments of the patient’s state proposed by the students, the decisions and interventions they make. It also enables communication between the student and the tutor, as shown in the figure below (Figure 3).

The system enables the creation of ad hoc websites which provide support services.

The list of patients displays a registry of patients entered into the system. For a given user (employee or student), only the list of these patients who were assigned to them from an administrative level is displayed. The list is divided into categories (current patients, discharged patients and a registry of teaching documentation). Moreover, it enables rapid and quick searching for a patient based on their personal data or their location in the hospital (by selecting a ward). In the list, in addition to the basic data, one can also see (in the form of graphic icons) information concerning the data missing on the patient’s medical history chart, and also a list of assessments of systems which were already performed for the patient. If at a given time an intervention is planned for the patient in question (an activity resulting from the care plan), the patient is highlighted on the list above by an exclamation mark displayed next to their data record. An “Individual Nursing Care Chart” is assigned to each patient, based on the eHealth Council Recommendation, in which the required elements of a physical examination are provided.

In subsequent stages of the work on the interoperability of data in the application, the medical history was mapped to a reference terminology (the terms found in the classification were compared to the terms from the terminology), and the results of the work were published, demonstrating the ability to use the dictionary in all elements of patients’ description in the application [3].

Another element of the nursing care process in the application are patient assessments. From the level of this module, the employee enters data concerning the assessment of the patient’s health status, using the terminology implemented in the system, which allows the indication of the patient’s symptoms, problems and assessments using standardised care quality indicators C-HOBIC [51]. Based on the rules established in the administration panel (standardised care plans were prepared in subsequent stages of operation), the system may prompt certain interventions or suggest diagnoses which result from the data entered in the subsequent assessment sheets. The rules are flexible and enable advanced inferences, also based on the working notes of the employees. All assessments have version control, which enables showing subsequent assessments over time. This enables the operator to monitor the changes to the patient’s health status on an ongoing basis. For the description of the assessments, standardised *ICNP* terminology and previously listed nursing care quality indicators recommended by the International Council of Nurses should be used. The indicators were prepared through an analysis of millions of patient records by the Registered Nurse’s Association of Ontario, in a project financed by the Minister for Health in Ontario (Canada). This system was called C-HOBIC [51]. The ADPIECare system contains 10 recommended indicators.

The person working with the application may move the screen down in order to select the system assessed in the patient or use an extended panel to move to the assessed system in Figure 4.

The next module is the “Diagnoses and care plans” module, which supports the operator in developing a care plan for the patient by selecting from the diagnoses suggested by the system based on the patient’s health status assessments entered in the system. Every diagnosis selected by the operator becomes a separate care plan, for which the operator may plan the next interventions (activities) at specific times and with the use of a device necessary to perform the activity in question. After a diagnosis selection decision is made, a care plan is created, supplementing it with proposed nursing interventions or with the option of searching for interventions in the entire terminology. Creating interventions and establishing their frequency (realisation, execution and commencement time) is another system functionality. To each care plan, a text description may be added. The care plan ends with an evaluation by selecting a diagnosis, positive, negative or a diagnosis from a risk group, which are appropriately grouped in the International Classification for Nursing Practice—*ICNP*. The information documented by the nurse is used to create a nursing report, in which the last record of patient activity is visible, enabling the continuation of care at another place.

Moreover, the application supports the user by enabling selection from finished care plans of those plans which are adequate to the assessment conducted. The student/nurse may choose from among the proposed interventions the ones that are possible and for the performance of which no competencies are unavailable. The system monitors and shows the impending interventions using an “Alert” icon. The possibility of adding interventions to the diagnosis is shown by an arrow on the figure below. Every care plan ends with an evaluation (assessment of the plan), consisting of making a final diagnosis from the categories of positive, negative, or at risk. Evaluation is a result of the care plan implemented in practice and the final diagnosis is used for continuation of care—the creation of another plan, which is adopted by the nurse to the patient’s current situation. The system provides support in establishing the care plan with its ICNP terminology standardised care plans. The application contains care plan standards which were established in cooperation with students and nurses who participated in care plan planning training courses using the ADPIECare system. An example care plan view, showing a set of terms concerning the intervention which the nurse may adapt to the patient’s health status, is shown in the figure below (Figure 5).

When working with the application (2017–2021), the users (students and nurses participating in the training) prepared care plans for 577 diagnoses. Figure 6 shows the diagnoses selected by students from the range of over 100 care plans for a given diagnosis. Nursing diagnoses have value for predictive models which demonstrate the risk to health and life of the patient, regardless of medical data. 

The next module in the user panel is “Data analysis”—a module which enables analysing the changes to the patient’s health status over time, based on the assessments entered into the system. The system enables the creation of dynamic reports based on a single variable or on multiple variables. Based on the numeric data or from the C-HOBC assessment scale, the system natively generates a linear or star (radar, spider) chart, as shown in Figure 7.

## 2. Materials and Methods

A discharge chart is a module which enables the assessment of the readiness of the patient for discharge (checking their knowledge concerning the medication they are taking or their illnesses) based on the C-HOBIC tool. The discharge report may be printed from the system in order to be handed to a patient and included in the patient’s medical documentation. The possibility of printing out ensures continuity and safety of care to the patient and the carers.

The goal of this study is to establish the usability of the original “ADPIECare Dorothea” application used for the management of documentation.

This study was conducted at the Medical University in Łódź among the users of the ADPIECare “Dorothea” application, 78 nurses studying for an MA in nursing. This study was performed during seminars of the “European Nursing” course. The study group was made up of nursing students in the second year of the master’s programme in Nursing, the Health Sciences Department. The eligibility criterion for a given group to participate in the study was previously completed classes, during which students received instructions on the use of ADPIECare Application and the principles of planning of care using the ICNP reference terminology. The courses were conducted under the supervision of the software’s author, using computers which allow working with the application. At the first stage, the design of the tool and the sequence for filling out data were presented. The composition of the ICNP nursing classification used in the tool was explained and the C-HOBIC indicators referring to the results of the nursing care quality for the needs of assessment of self-management of care by the patients and preparation of interim care plans were discussed. After initial discussions of the principles of work, the students were divided into groups. Then, they received a case description, based on which they filled out individual elements of electronic documentation, a medical history chart, assessment of systems and prepared a care plan, composed of an initial diagnosis resulting from the assessment along with the interventions. At the final stage, the students’ task was to conduct evaluation by indicating a final diagnosis/result of care in the plan using ICNP terminology. In the first step, students performed an assessment, and then they made a diagnosis, selected an intervention, and in the last step of the process they conducted an evaluation for the case description prepared for the examination.

The medical history chart is the basic element of the electronic documentation system, which is the foundation of patient assessment. Filling it out enables work in subsequent steps in the application. A key resource is the body systems, according to which the assessment chart was created. The students had a task of indicating, in a specially prepared form, the elements they identified for assessment in the system in question. Thus, we understand the “system” to mean elements of assessment concerning, for example, the respiratory system, the circulatory system, the gastrointestinal system, etc.

The care plan concerned the selection of diagnoses which meet the patient’s health status and selecting appropriate interventions intended to meet the patient’s needs, that is, achieving a positive diagnosis during the evaluation of the care process. After the courses were finished, students were requested to fill out a survey questionnaire assessing the usability of the system. The study was conducted in the period of November–December 2019. The survey questionnaire was sent by electronic communication to a group of students participating in the research process. To create the survey questionnaire, Google Forms were used; where variables specified in a standardised *Defining and Testing EMR Usability MASTER V2 Final Healthcare Information and Management Systems Society* (*HIMSS)* tool were entered, a post-usability test was used [29]. A common approach is to use Likert-scale questionnaires, where users are asked to rate their satisfaction with various aspects of the product; hence, this study used a Likert scale of 1 to 5, where 1 means strongly disagree and 5 means strongly agree (Appendix A). From the obtained results, the mean of the responses (62 responses) to individual variables was calculated. Additionally, open-ended questions were added, used to assess which screens were user-friendly and which not. Users could indicate which elements of the system were easy or difficult to operate. Afterwards, the tool was made available to the students on the Facebook social networking site in a group of 2nd year students which participated in the study (78 studying nurses were in this group) and 62 answered the study. The students created a private group for the year, which enabled using this form of communication, with Facebook Messenger used as the group’s communicator for consultation. The tested group was larger than the group recommended by system usability researchers, who indicate that an assessment by 5 users is a sufficient number [29].

## 3. Results

The survey of nurses’ opinions on the usability of IT systems was commenced by checking when the “ADPIECare Dorothea” application was used for the first time (Figure 8). As the received answers show, the majority of people had contact with the application during the first year of the master’s programme. Only a small number of respondents have encountered this tool during their education in the second year of undergraduate studies. As the existing situation demonstrates, students did not encounter the application during the remaining years, although most of them should come across the system during their lessons in the first year of study (Figure 8). 

The results of the research indicate that most users who are in contact with the application for documenting medical events have used it 2 to 5 times, which constitutes 70% of all research subjects. A smaller group, constituting 26% of the subjects, was the group of respondents who used the application only once. Only 4% of the surveyed persons entered information in the application from 5 to 10 times (Figure 9).

The study demonstrates severe differences in the frequency of use of the application during testing. Only a few users could demonstrate the highest number of attempts; however, there was not a single person who did not operate the system on their own (Figure 9).

As the answers in Figure 10 demonstrate, as many as 80% of the respondents did not have any contact with the application outside of the training course. Much fewer people, only 17%, used the tool for their own purposes from 5 to 10 times; 3% of people used it only once (Figure 10).

Another cross-section of information about the surveyed persons concerns the goal of the use of the application during its first use. The data presented in the figure above demonstrate that users most frequently filled out the medical history chart. “Assessment of body systems from 1 to 2” was filled out by five persons. At the “Assessment of body systems from 2 to 5”, this value increased twofold. Six persons filled out both “Assessment of body systems more than 5 times” and “discharge readiness scale”. “Pressure ulcers” were filled out by three persons. Care plans using diagnoses and interventions were filled out by the following number of persons: “1 care plan”—five persons; “2 to 5 care plans”—eight persons. The time consumed by the use of the application when preparing a care plan was analysed. The answers of the respondents enable establishing that preparing a care plan for a single patient most frequently takes from 21 to 40 min. Slightly fewer (seven) respondents filled out a care plan in a time interval of 11 to 20 min. The most experienced persons only needed 10 min. An analysis of the chart shows that the least numerous group consists of people who marked an answer of 41 to 60 min (Figure 11).

The assessment of the user interface (system had a clear, easy to understand and ordered appearance of the screen) of the ADPIECare system is shown below; the answers “strongly agree” and “mostly agree” amount to a total of 70%. This demonstrates that most of the surveyed persons considered the application screen to be clear, easy to understand and ordered. A total of 26% of respondents selected the convenient option of “I do not know”. Additionally, only 4% expressed their disapproval by selecting the answer “mostly disagree”. None of the nurses selected the answer “definitely disagree” (Figure 12). 

The feelings of students regarding the usefulness of the ADPIECare “Dorothea” system were generally positive and only for the variables (2) “The application kept screen changes to a minimum during completion of a task” (Mean = 3.4) and (3) “The application minimised the number of steps it took to complete tasks” (Mean = 3.4) the mean was at the neutral level, while in the case of the first variable, 17% of the respondents indicated that they did not have an opinion in this respect, and in the case of the second variable it was as much as 30%. In the case of other variables such as (1) “ The application had clear, clean, uncluttered screen design” and (6) “Choice lists were clear and unambiguous” (Mean = 3.5); (7) “Clinical documentation tools were efficient to use” and (9) “Data could be entered once then used in multiple places” (Mean = 3.6); (4) “Information presented on screens was easy to comprehend quickly” and (8) “Alerts were only presented at appropriate time” (Mean = 3.8); and the last question (5) “Information needed for a specific task was grouped together on a single screen” achieved a Mean score of 3.9. In the case of the above-mentioned variables, 48% of respondents felt (Strongly agree and Mostly agree) that “The application minimised the number of steps it took to complete tasks” and “Alerts were only presented at appropriate time”; 56%—“Clinical documentation tools were efficient to use”; and 70%—“The application had clear, clean, uncluttered screen design” and “Information presented on screens was easy to comprehend quickly”. Almost 3/4 of the respondents felt that “Data could be entered once then used in multiple places” and “The application kept screen changes to a minimum during completion of a task”. Every second respondent (81%) felt that “Information needed for a specific task was grouped together on a single screen”. A total of 19% of nurses (students) do not have an opinion on this matter. None of the students had any negative feelings in this case. Negative (22%) and neutral (30%) feelings concerned the variable “The application minimised the number of steps it took to complete tasks” and positive feelings (Mostly agree and Strongly agree) were indicated by 48% of nurses assessing the usefulness of the system.

Based on the answers of the respondents, it may be established that the elements which created the most difficulties when working with the application were: “selection of diagnoses” (15 persons), “selection of interventions” (14 persons) and “planning the work pattern” (15 persons). The same result (7 persons) was found for both “assessment of systems” and to “using tools to assess the patient’s health status”. Slightly less, six persons, had problems with preparing a discharge readiness assessment. The interview card proved to be the simplest to fill out.

When analysing the chart, the “medical history chart” proved to be the simplest to use, marked by 16 persons. Next places were, in order, “assessment of systems” (marked by 14 persons) and “using chart, tools to assess the health status” (Figure 13).

There were only eight persons for whom “assessment of systems” was easy to use, six persons who selected the answer “preparation of discharge readiness assessment” and five persons chose the “selection of intervention” answer. The lowest score was obtained for the “planning the work pattern” answer (Figure 13).

Despite little experience with the ADPIECare Dorothea application, over half of the respondents had a positive attitude towards the planning of patient care with the use of the application. A total of 30% were reluctant with regard to the introduction of the system. Only 9% did not have an opinion on the matter, while 4% were hesitant and stipulated acceptance on the condition that changes are made to the category of selecting of diagnoses and interventions (Figure 14).

The last question in the questionnaire was an open question. It allowed us to thoroughly learn the opinions of the nurses about the application, and also the justification of their choice in item 17. A significant majority of the answers concerned positive comments. An opinion survey is always related to the disapproval of the surveyed parties, which was also reflected in this case. A small number of persons were cautious and had no opinion concerning the IT tool (Table 1).

## 4. Discussion

The presented study and the conducted surveys were inspired by current trends of digital transformation in health care. The e-Health revolution uses IT and telecommunications technologies for the development of the medical sector in Poland, and also in other European Union countries [52]. The selection of the studied group was not accidental. Most of the surveyed persons were second year students, who were being prepared during their education to work in a digitised world. Additionally, they are characterised by a young age, less work experience, and a fresh approach to the currently introduced standards of modern nursing care.

The nature of information and communication services fosters the dissemination of knowledge by separating data from the physical location of the problem. Geographical boundaries are ceasing to exist and allow the integration of distant societies, which has a positive impact on the general availability of knowledge for everyone [53]. It is important to approach the issue of electronic documentation and information in a comprehensive and interdisciplinary manner and this system can be used to develop interdisciplinary care plans [52].

Another study also demonstrates that nurses do not yet have the knowledge nor opinions about the implementation of electronic documentation. The conclusions of the referred study refer strictly to the need to create a new course concerning nursing IT, and also to undertake efforts to educate students in the documentation of medical events, with the help of the teaching staff. It is possible that such an initiative could help in the more effective use of the tool and in the understanding of the process related to general digitisation [54]. Currently, the students undergo the process of training with documentation tools at the first level of graduate education, as part of the “primary health care” course. These changes are reflected in the outcomes of this study. The respondents declare closer contact with the application and positive approach to working with it.

The aforementioned analysis of survey studies, in the answer to the question “Do nurses have a positive attitude to the daily work using the ADPIECare Dorothea system”, allows establishing that the positive approach of the nurses to patient care planning using the system may result from presenting a method for improving the documentation of medical events by the nurses. To justify the respondents’ willingness to use the “ADPIECare Dorothea” system, a table of results was created (Table 1). In this table, the answers were divided by the researcher into three subjective categories: positive, neutral and negative. As demonstrated by the analysis, an overwhelming number of arguments refers to the concept of an IT tool, emphasising its clear and modern look. The time savings, which are of key importance in the nursing profession, were also marked as an advantageous element. Nurses are burdened daily with many medical and administrative activities. Effective data input into the IT system, and also access to these data, would enable comprehensive and more effective patient care [55]. Having appropriate collected data at one’s disposal enables appropriate decisions at a correct time [56]. The issue of data exchange [57] and the possibilities of individual applications also have to be emphasised [58,59].

Referring to the right side of the table, where negative answers were shown, it can be established that digitisation of medical documentation causes some anxieties among its users. It may result from attachment to paper documentation, or from the time needed to fill out the documents [60,61]. By analysing the counterarguments, the deficiencies of hardware and access to more modern technology are highlighted. Maybe it results from financial limitations of the public health care entities, but another possibility is also not taking into account this professional group when the tool for nurses was being designed [62].

The interest of the researchers focused on the assessment of the view of the screen and of the legibility of information presented in the application. We can obtain this answer when simultaneously analysing the results of the sixth and ninth questions (6. The system had a clear, easy to understand and ordered appearance of the screen; 9. The information on the screens was easy to understand). The percentage of respondents with a positive opinion was relatively high. The legibility of functions is of key importance in the skilful use of the application. Its chaotic appearance, the layout of the contents or terminology is incomprehensible to the user, could have an adverse impact on the decision regarding repeated use.

An important element for the assessment of the use of IT tools is the time needed to fill out documentation. Usually, IT systems have multiple functionalities and a nurse does not have to use every functionality. This all depends on the patient’s condition. This is why it is so important whether the documentation of a single patient care plan was time-consuming. The answers can be obtained by analysing the results of the fifth question of the survey (5. The time spent to document a care plan for a single patient). The question was intended to indicate the amount of time necessary in order to document a care plan for a single patient. The surveyed person could choose from five answers, which enabled establishing the time needed for work only approximately. Most nurses used from 21 to 40 min to prepare the documentation. However, to precisely establish the time needed for the documentation process the research should be expanded by individual average time of filling out data, with the same number of approaches to the task. It should also be noted that a significant factor which impacts the results is the experience of each of the surveyed persons and the characteristics of the patient’s problem. To summarise the studied problem, it may be established that preparing a single care plan is time-consuming. It is possible that the ability to use the classification and the knowledge of used terms which are not commonly used may have an impact on this. This lengthens the time needed to use the system at this stage of working with electronic documentation. Therefore, if it was not possible to acquaint oneself with the classification earlier, searching through a database of over eight hundred diagnoses may have an impact on the time needed to fill out the documents and be reflected in the opinion of the users.

To recapitulate, the application follows the current trend of coordinating the IT department with the medical department. The implementation of decision-supporting systems with the use of terminology (enabling reliable and uniform documentation) may provide the patient with a better quality of medical services and continuity of care, due to access to full nursing information. A useful feature of this tool is the use of prepared care plans based on the ICNP^®^ international classification. The universality of the language for the coding of diagnoses and interventions enables transmitting the data in an electronic format also to locations outside of Poland.

## 5. Conclusions

In this paper, we have presented a system assessment particularly from the perspective of the end users’ interactions. Conventional methods of assessing health information systems have limitations and can benefit by complementing them with open-ended questions, indicating what factors are difficult or easy to use.

The medical history chart proved to be the simplest to fill out [63], which may indicate that the frequency with which this intervention is performed during the studies is higher, whereas the remaining phases of the nursing process based on ICNP terminology are insufficiently memorised. Moreover, insufficient practical knowledge in the use of reference terminology and key words may prove to be a hindrance and impact the assessment of the use of the system.

The system can be used in practice if it meets the expectations of users and is adapted to their needs. Although the respondents indicated that the preparation of care plans, the selection of ICNP interventions and nursing diagnoses were the most difficult, the respondents’ feelings were positive and inclined to accept the use of this solution in practice. It took the longest time to develop the care plans. Hence, in education, we should devote more time to teaching the use of the ICNP dictionary.

The implementation of the care planning system for teaching not only develops the digital competences of nurses, but also teaches them what elements the system can contain, and what is missing in it. Thanks to the problems visible in the system, nurses can point out elements that are important to them in designing the system architecture.

## Figures and Tables

**Figure 1 ijerph-19-08805-f001:**
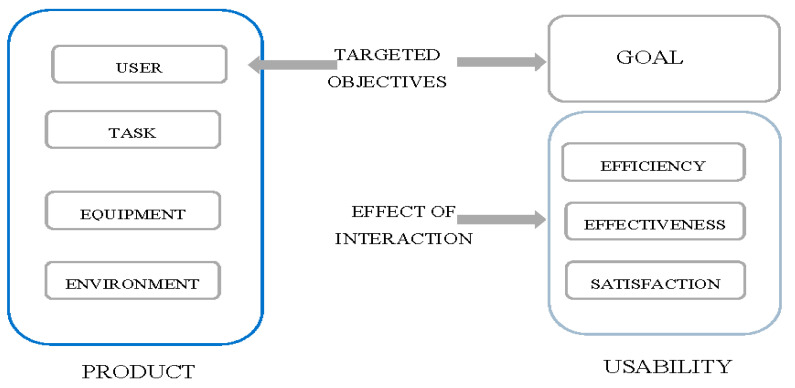
Achieving usability in an interactive product improvement cycle. Source: M. Kusiak, *Usability of information system in the opinion of nurses* (Master thesis) Medical University of Lodz, 2020.

**Figure 2 ijerph-19-08805-f002:**
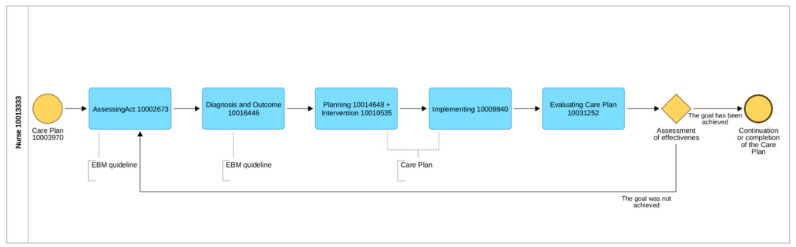
The BMN clinical path—the nursing process in client/patient education. Source: BPMN ed. Kamiński M., Kilańska D., Lipiński C., Librowska B., Szydłowska-Pawlak P., Dział Rozwoju Systemów Opieki Zdrowotnej (Department of Health Care Development Systems), ADONIS, UMED 2020 (for AOTMIT 2020).

**Figure 3 ijerph-19-08805-f003:**
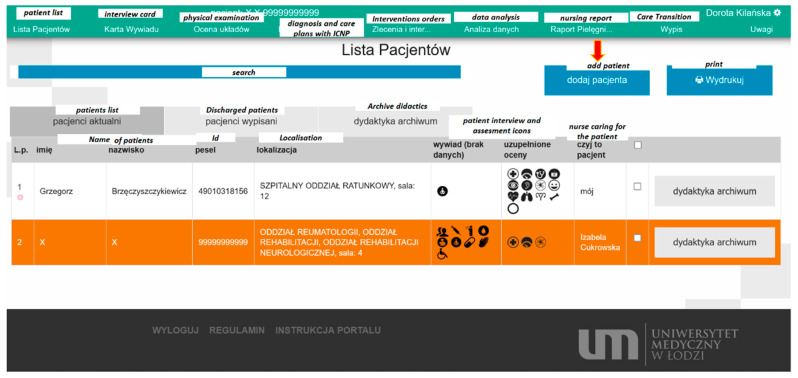
The ADPIECare “Dorothea” system’s user panel; source: https://pielegniarki.umed.pl/lista_pacjentow.html (accessed on on 19 April 2022).

**Figure 4 ijerph-19-08805-f004:**
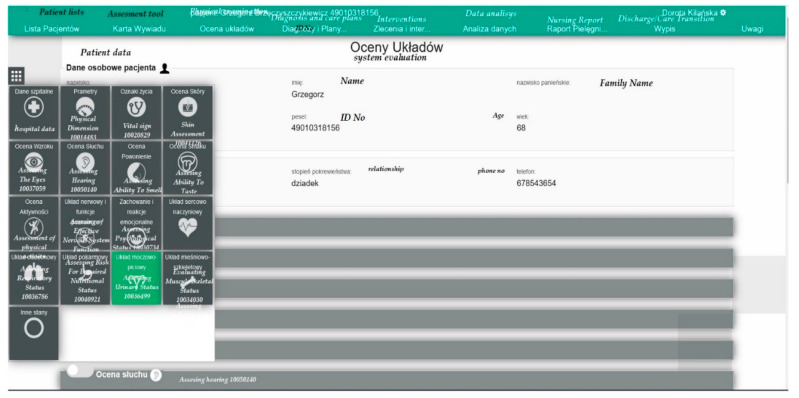
Iconography of physical examination in ADPIECare “Dorothea” using ICNP^TM^ print screen; source: https://pielegniarki.umed.pl/oceny.html (accessed on on 19 April 2022).

**Figure 5 ijerph-19-08805-f005:**
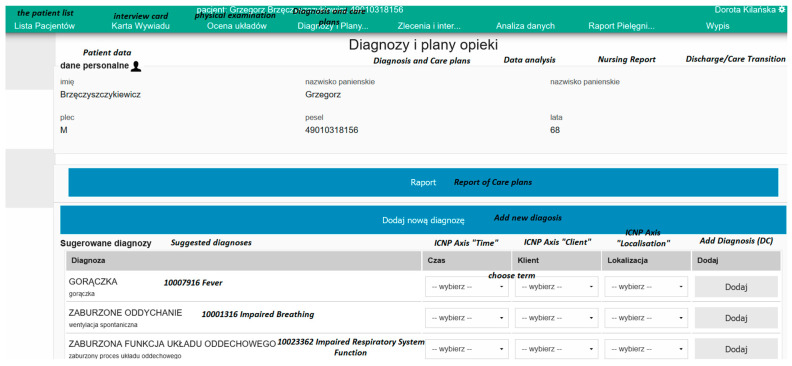
An example set of diagnoses for a patient enabling establishing the time, location of the problem; source: https://pielegniarki.umed.pl/diagnozy.html (accessed on on 19 April 2022).

**Figure 6 ijerph-19-08805-f006:**
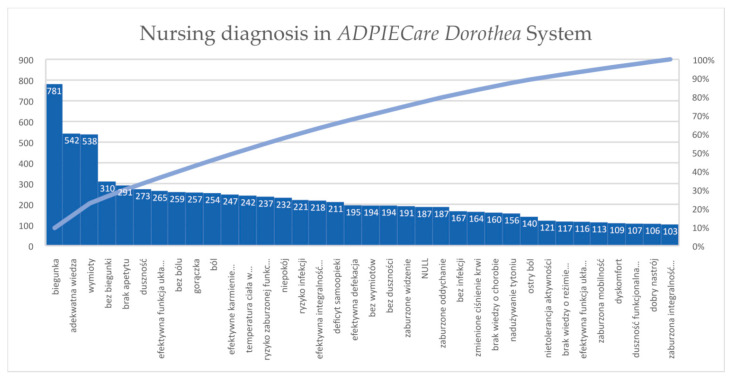
The most frequently selected nursing diagnoses assessing patients in ADPIECare Dorothea made by students in the years 2017–2021 from a range exceeding 100 care plans for a given diagnosis.

**Figure 7 ijerph-19-08805-f007:**
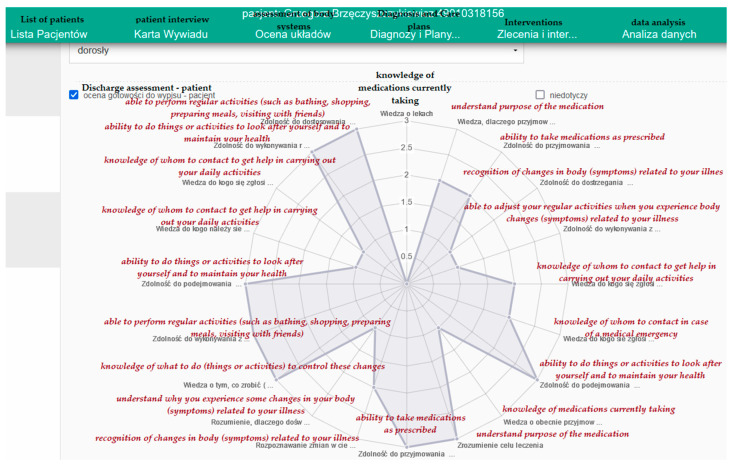
Assessment of the patient’s health competences with the use of C-HOBIC; source: https://pielegniarki.umed.pl/wypis.html (accessed on on 19 April 2022) [52].

**Figure 8 ijerph-19-08805-f008:**
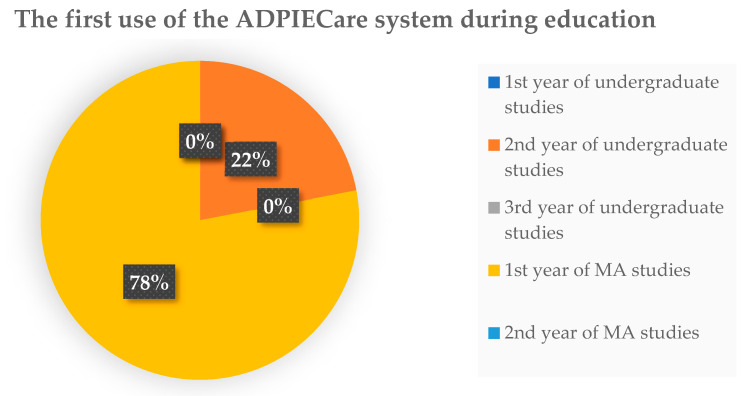
Experience in the work with ADPIECare system application by the surveyed users during undergraduate (BSN) and MA studies (MZ).

**Figure 9 ijerph-19-08805-f009:**
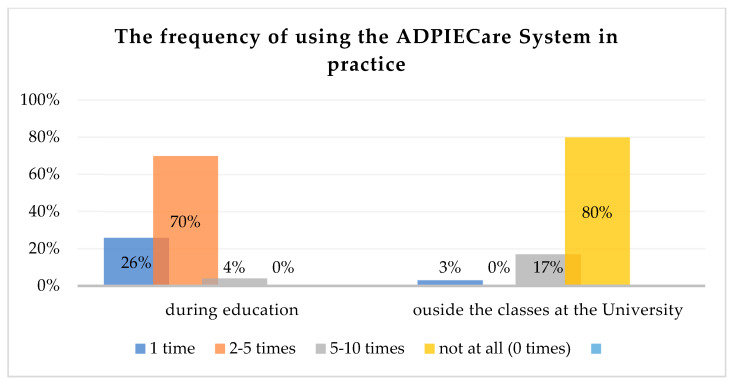
Frequency of using the ADPIECare system.

**Figure 10 ijerph-19-08805-f010:**
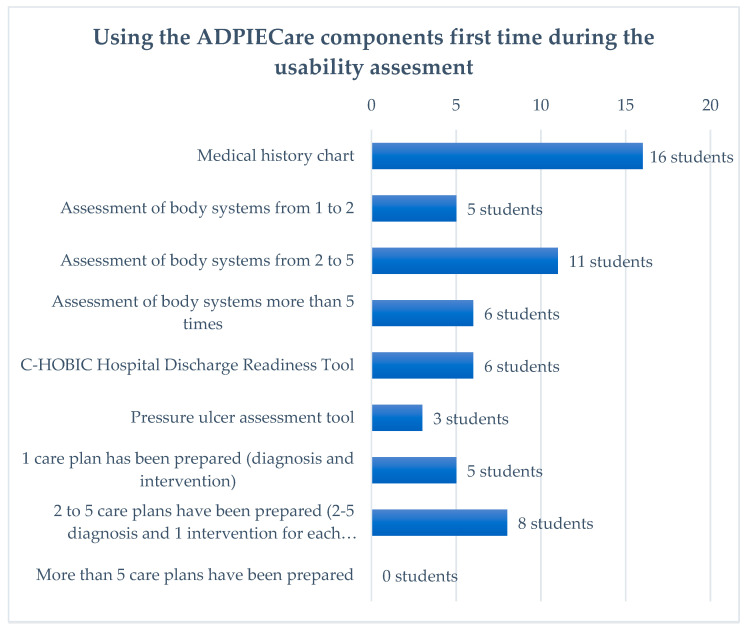
The use of individual components of the ADPIECare system by the surveyed students.

**Figure 11 ijerph-19-08805-f011:**
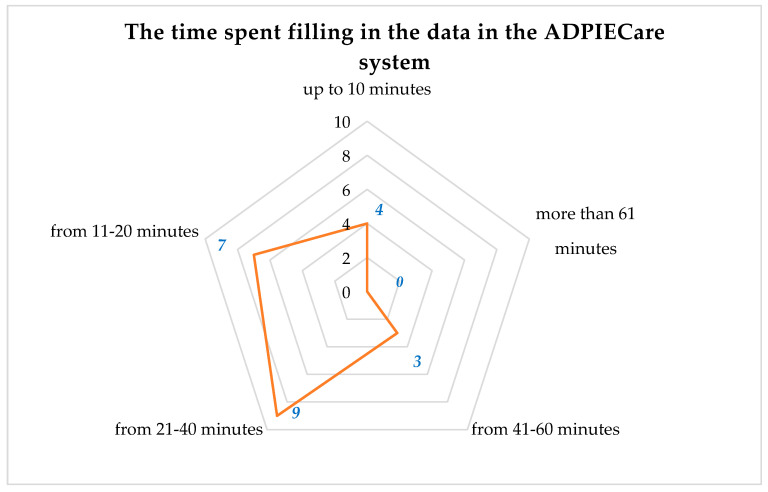
Time consumed by documenting the patient’s care plan.

**Figure 12 ijerph-19-08805-f012:**
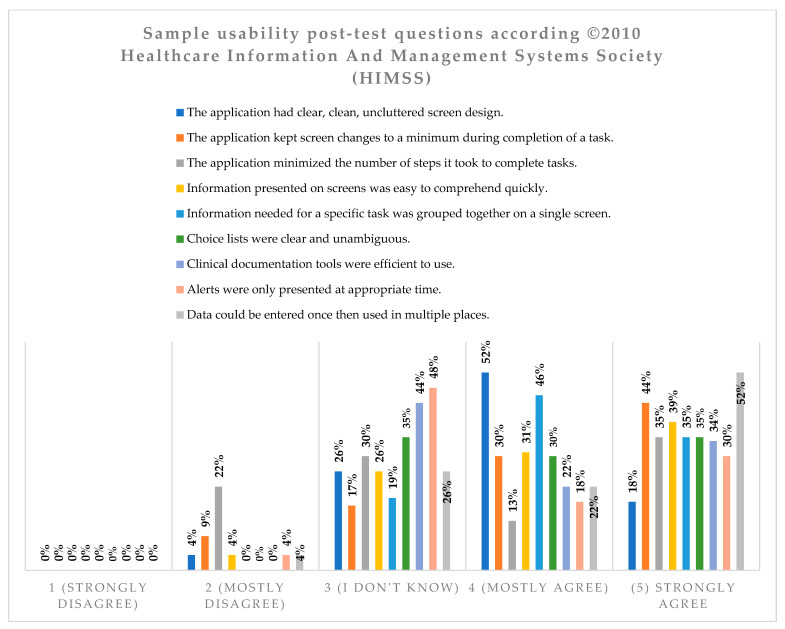
Characteristics of the ADPIECare system user interface. The assessment of ADPIECare system by nurses.

**Figure 13 ijerph-19-08805-f013:**
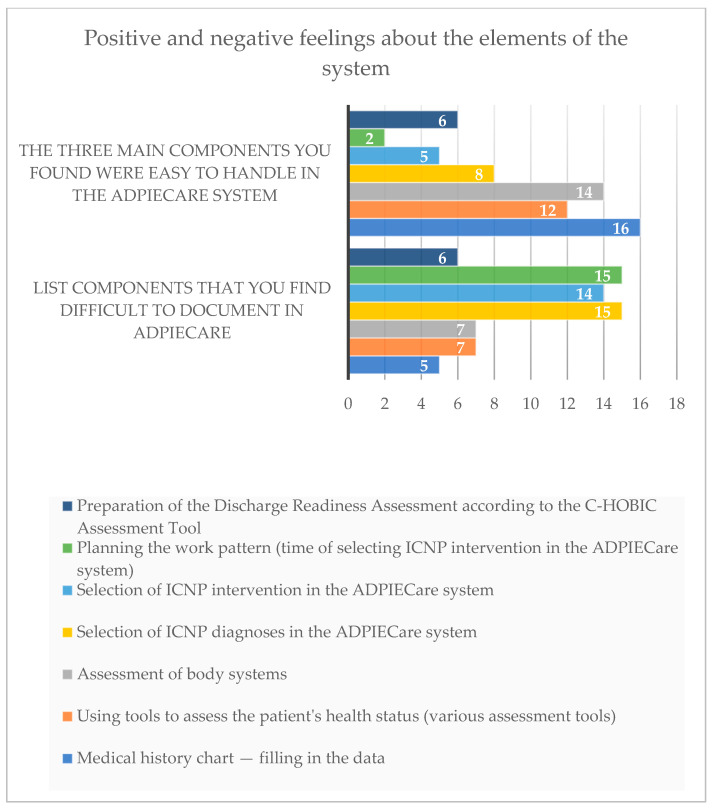
Factors that cause problems in the documentation of data in the ADPIECare system in the opinion of the surveyed users of the system.

**Figure 14 ijerph-19-08805-f014:**
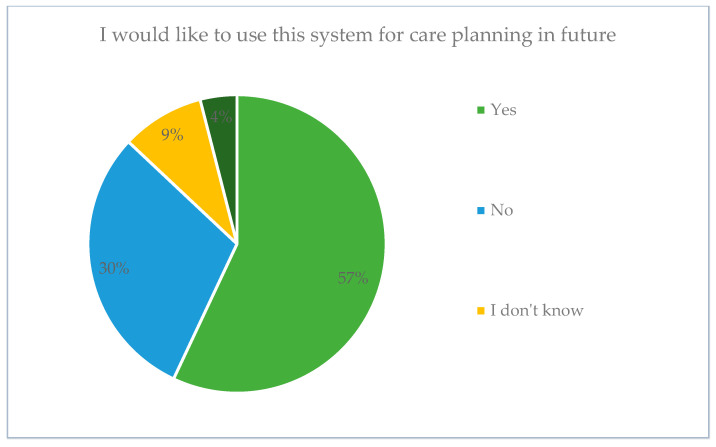
Attitude of the respondents to the implementation of the application for patient care planning.

**Table 1 ijerph-19-08805-t001:** Responses of the surveyed to question 18 from the survey.

Positive Answers	Neutral Answers	Negative Answers
the application is intuitive the application facilitates work all patient assessments can be found in a single location the time needed to input data would be shortened, even though such detailed assessment of individual systems is not necessary yes, if the option of selecting diagnoses and interventions in the system is improved everything is legible and comprehensible the application is state-of-the-art it facilitates the collection of necessary data rapid collection of data about the patient the software would facilitate and accelerate care planning and would group patient information yes, if this will have a positive impact on better wages	no opinion application has both advantages and disadvantages I don’t know	too much time too much filling out anxiety about server or computer freezing and insufficient number of computers at the ward I don’t need this in the daily work the software narrows the manner of formulating of diagnoses and interventions, sometimes it is difficult to adapt the proposed description to the actual problem I prefer paper documentation

## Data Availability

Non-digital data supporting this study were curated by Dorota Kilanska.

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
