# Peer review of "The Usability of IT Systems in Document Management, Using the Example of the ADPIECare Dorothea Documentation and Nurse Support System"

_ijerph, 2022, doi:10.3390/ijerph19148805_

Round 1

Reviewer 1 Report

This paper provides results of the study of the usability of a developed IT system in document management for the nurses. Overall, the paper is interesting and well written but contains required improvements prior to considering it for publication.

1. Introduction section should be significantly improved in terms of "storytelling". Most paragraphs and sentences are too long and provide a lot of information, including references, without a clear structure. An example of an unclear long sentence is the sentence in lines 39-45. It would benefit the reader to have a better structure of the presented background and related work. Minor typos in this section: line 98 ("As much as ...", not "has"); line 117 ("how easy it is to perform ...").

2. Section 1.1. starts with a quote that does not have a reference. Is it a quote from the work or just an opinion of the person in question? Why is this part of the text written in italic?

3. Appendix 1 mentioned in line 249 presents the entire Gantt chart of the project, which is not typical for scientific papers. What is the relevance of the Gantt chart for making the conclusions about the hypotheses presented in this paper? Please elaborate on the importance of the chart or remove it.

4. Some figures should be improved, e.g. Fig. 3-5 have overlapping text.

5. Please rephrase the sentence in line 460. Here you mention that students did not encounter the application during the remaining years. Does that mean they used it only in their first and second year of the studies, or do you mean that no students used it for the first time after the 2nd year?

6. Figure 10 - what does "at all" mean on the chart?

7. Figure 11 - there seems to be an overlap between two items: "assessment of body systems from 1 to 2" and "assessment of body systems from 2 to 5". What are those body systems? The discussion around the figure does not mention the types of body systems.

8. One question in the analysis is strange: "During use, the application kept the introduced changes". What does it mean? If the application does not save the input data, it is not useful as a data management system. More importantly, if some users reported that it does not keep the introduced changes, it means that they discovered a bug in the system which prevents it from saving data. Was it addressed?

Author Response

authors

Reviewer 2 Report

Thank you for your manuscript, however in its current form it resembles more a university study report, than a scientific publication. A number of recommendations are provided below:

Major points:

- Line 54. This is not correct. The existence of EMR is not synonymous with the improvement of care. It is only a means to an end, and any improvement is dependent on the use of the EMR.

- The introduction lacks cohesion in the arguments listed, and as such consists of a large number of unlinked statements. For example, lines 51-63 constitute a sequence of unlinked statements. Please rephrase into a coherent set of arguments, where there is a clear causal relationship and purpose.

- It is not clear why there is a entire section copied in 1.1, instead of a more succinct presentation of the system.

- There is a very extensive description of the IT system. This is not appropriate for the journal, please provide a succinct description of the IT solution, and if you want a more extended version can be provided as supplementary material. Otherwise the flow of your manuscript is severely disrupted.

- In methodology, it is not stated how many students participated in the facebook group- and if all were active during the time period (which is also not specified exactly).

- in the methodology, please provide the questionnaire as supplementary material.

- Figures 8-10 can be combined in a single figure 

- In the methodology, why was this system only tested with student nurses - and not included practicing nurses - the latter would presumably provide a more pertinent feedback for usability and eventual deployment in the field.

- Figures 13-21 can be summarised into a single table.

- Lines 633-635, that is correct, but the current study is not inter-disciplinary

- The hypotheses are clearly articulated in lines 700-712, this should happen in the introduction

- While a significant number of respondents raised questions or were uncomfortable with the electronic system, this aspect is not considered carefully, but only briefly mentioned in lines 662-669.

- The conclusion section is inappropriate. Conclusions should summarise the study and place it within a given context in relation to current and future events. Your conclusions repeat information that has already been provided.

Minor points:

- Sentences are often too long and can lead to confusion for the readership. E.g., lines 39-45, 

- Lines 46-50, that is to be expected, not sure why it merits a specific mention as it does not link with the subsequent argument clearly.

Author Response

authors

Round 2

Reviewer 2 Report

Thank you for your responses. A few minor comments to be addressed, as per below:

- The following section is long-winded, it can be further compacted. From "A measurement of safety...." to "... and monitoring (1%)"

- Why are the 5 Nielsen principles presented in such detail, if your discussion does not address them one by one? A simple mention would suffice.

- The section "In the source literature..." to "...key importance to fill this gap" is not referring to Poland, while subsequently Poland is specifically mentioned. As this study is relevant to Polish initiatives and the situation in Poland, the EU context can be compacted to fewer sentences (the references are appropriate).

- Reference 66 is incomplete.

Author Response

Thank you very much for your help in increasing its value and any suggestions that helped to organize the content.

Finally, we shortened the section indicated "the section is long-winded, it can be further compacted. From" A measurement of safety .... "to" ... and monitoring (1%) "and completed the section" In the source literature ... "to" ... key importance to fill this gap ", showing the importance for the context of Polish health care system;

We also limited 5 Nielsen components to the minimum necessary information what it consists of and  analyzed the literature and supplemented the reference 66.

Additionally, as suggested, the article was analyzed by a native speaker and the final version with the above-mentioned changes was attached to MDPI.

Thank you very much for your help and time devoted to analyzing the publication; Thanks to careful analysis, we have learned a lot and it will be helpful in future publications for which we are very grateful. Thank you for your consideration of this manuscript.
